# Non-IgE-Mediated Gastrointestinal Food Allergies in Children: An Update

**DOI:** 10.3390/nu12072086

**Published:** 2020-07-14

**Authors:** Roxane Labrosse, François Graham, Jean-Christoph Caubet

**Affiliations:** 1Division of Hematology-Oncology, Department of Pediatrics, Boston Children’s Hospital, Boston, MA 02115, USA; roxane.labrosse@childrens.harvard.edu; 2Division of Allergy and Immunology, Department of Pediatrics, CHU Sainte-Justine, University of Montreal, Montreal, QC H3T 1C5, Canada; francois.graham@umontreal.ca; 3Division of Allergy and Immunology, Department of Medicine, Centre Hospitalier de l’Universite de Montreal (CHUM), University of Montreal, Montreal, QC H2X 3E4, Canada; 4Pediatric Allergy Unit, Department of Woman, Child and Adolescent, University Hospitals of Geneva, 1205 Geneva, Switzerland

**Keywords:** food allergy, non-IgE-mediated, nutrition, pediatrics, FPIES, FPE, FPIAP, gastrointestinal reactions

## Abstract

Non-immunoglobulin E-mediated gastrointestinal food allergic disorders (non-IgE-GI-FA) include food protein-induced enterocolitis syndrome (FPIES), food protein-induced enteropathy (FPE) and food protein-induced allergic proctocolitis (FPIAP), which present with symptoms of variable severity, affecting the gastrointestinal tract in response to specific dietary antigens. The diagnosis of non-IgE-GI-FA is made clinically, and relies on a constellation of typical symptoms that improve upon removal of the culprit food. When possible, food reintroduction should be attempted, with the documentation of symptoms relapse to establish a conclusive diagnosis. Management includes dietary avoidance, nutritional counselling, and supportive measures in the case of accidental exposure. The prognosis is generally favorable, with the majority of cases resolved before school age. Serial follow-up to establish whether the acquisition of tolerance has occurred is therefore essential in order to avoid unnecessary food restriction and potential consequent nutritional deficiencies. The purpose of this review is to delineate the distinctive clinical features of non-IgE-mediated food allergies presenting with gastrointestinal symptomatology, to summarize our current understanding of the pathogenesis driving these diseases, to discuss recent findings, and to address currents gaps in the knowledge, to guide future management opportunities.

## 1. Introduction

Gastrointestinal (GI) complaints represent a frequent motive for seeking medical attention in the pediatric setting, and diagnosis can be challenging due to the wide variety of potential underlying causes. Particularly in the first years of life, food allergies represent a substantial proportion of disorders involving the GI tract [1]. Food allergies comprise a spectrum of diseases that have in common the immunological reaction to specific dietary proteins, and the reproducibility of symptoms upon re-exposure [2,3]. This is in contrast to food intolerances, in which immunological mechanisms are not involved. Non-IgE-mediated gastrointestinal food allergic diseases (non-IgE-GI-FA), which are being increasingly recognized in children, consist of three main entities: food protein-induced enterocolitis syndrome (FPIES), food protein-induced enteropathy (FPE) and food protein-induced allergic proctocolitis (FPIAP). Despite their potential for serious reactions, both first and second-line health care providers report poor familiarity with these disorders [4,5], perhaps due to the many gaps in knowledge, which include unclear pathophysiology, a paucity of reliable diagnostic tools, and a lack of uniform management protocols. This review will focus on non-IgE-GI-FA in children, summarizing what is currently known, and highlighting the latest advances in the field.

## 2. Classification and Terminology

Adverse food reactions are divided into immune-mediated reactions (i.e., food allergy) and non-immune mediated reactions (i.e., food intolerances). The term ‘food allergy’ is used to designate an immune-mediated adverse reaction to food proteins. This includes IgE-mediated food allergy (IgE-FA), mixed IgE and non IgE-mediated food allergy, and non-IgE-GI-FA [6]. On the other hand, adverse non-immune mediated reactions that are not classified as food allergy include malabsorption due to enzyme deficiency (ex: lactase deficiency), reaction to toxic contaminants (ex: scombroid poisoning), and pharmacologic food components (ex: caffeine) among others [7], which are beyond the scope of this review.

Gastrointestinal food allergies are generally classified according to their underlying pathogenesis (Figure 1) [2,8]. In IgE-FA, reactions generally occur rapidly after the ingestion of the culprit food. Although isolated gastrointestinal symptoms can occur (termed “gastrointestinal anaphylaxis”), typically with egg, more often, these are accompanied by other features affecting the skin and mucosa (hives, angioedema), respiratory tract (cough, wheezing, nasal congestion) or cardiovascular system (hypotension), and can present as life-threatening anaphylaxis. Reactions in mixed IgE/non-IgE-mediated diseases, such as eosinophilic gastrointestinal diseases, are triggered by complex immunological mechanisms that only partially implicate IgE. Symptoms are dependent upon the affected organs and the extent of eosinophilic infiltration [8,9]. On the other end of the spectrum lie non-IgE-GI-FA, in which circulating food-specific IgE are typically absent. These include FPIES, FPE, and FPIAP. In contrast to IgE-FA, associated GI symptoms are usually delayed after exposure to foods, and can have a chronic presentation [8,10]. Although their underlying pathomechanism is still poorly elucidated, these entities may represent a continuum of disease, where the expression of symptoms and severity is dependent upon the affected segment of the gastrointestinal tract (Figure 2). FPIAP symptomatology is induced by the localized inflammation of the distal colon, causing hematochezia in otherwise well-appearing infants. FPE predominantly affects the small intestine, resulting in lower digestive manifestations such as malabsorption symptoms, potentially accompanied by a failure to thrive (FTT). Finally, FPIES can affect the entire gastrointestinal tract, predominantly causing symptoms of intractable emesis which can be severe enough to cause metabolic disturbances and hypovolemic shock. FPIES can be further classified according to the timing of symptoms (acute vs. chronic FPIES), the severity of clinical manifestations (mild, moderate, severe), the age of onset (early-onset, late-onset, adult FPIES), the type of triggering foods (cow’s milk/soy vs. solid foods), and the presence of food-specific IgE (sIgE) (atypical FPIES) (Figure 3), all of which are described in detail below.

## 3. Epidemiology

Of the non-IgE-GI-FA, FPIAP is certainly the most frequent, although the exact prevalence is not well established. A large study of an Israeli birth cohort found the overall prevalence of FPIAP to be relatively low, at 0.16% [11]. Conversely, a cumulative incidence of 17% was found in a recent prospective observational study of healthy newborns with pediatrician-diagnosed FPIAP and established evidence of gastrointestinal bleeding in the United States [12]. The absence of diagnosis confirmation by reintroduction and the inclusion of patients with occult blood possibly led to overdiagnosis in the latter study. However, the incidence remained high (7%), despite restricting the analysis to those with gross blood only, thus only partially explaining the 100-fold difference between both studies. In infants with visible rectal bleeding, FPIAP has been found to be causal in up to 60% of cases [13,14].

The prevalence of FPE has not been thoroughly studied, but it is thought to be relatively uncommon, accounting for about a fifth of patients compared to celiac disease in reports from a single-center experience from Finland [15,16]. In another Finnish study, the prevalence of FPE to cow’s milk in older children was rather high, at 2.2% [17]. However, the incidence of this condition appears to be progressively decreasing over time [16,18]. Potential explanations for this decline include the upsurge in breastfeeding practices, which may be protective, and the use of better adapted formulas with a lower protein content [16].

While FPIES is also usually regarded as a rare disease, recent reports challenged this notion, revealing a cumulative incidence of 0.3–0.7% in infancy [19,20], although a lower incidence of 0.015% cases per year was also reported in Australia [21]. However, the latter study may have underestimated the incidence of FPIES, because it relied on a voluntary reporting system from pediatricians, and only included cases of acute FPIES. In cow’s milk-induced FPIES specifically, two independent studies conducted in Israel and Spain have both found comparable cumulative incidences in children during the first 2–3 years of life (0.34% and 0.35%, respectively) [19,20]. Similarly, the latest prevalence studies have estimated that 0.51% of children and 0.22% of adults in the United States report having physician-diagnosed FPIES [22].

Co-atopic disease is highly overrepresented in patients with non-IgE-GI-FA, affecting 40–60% of FPIES patients [23,24,25], and up to 40–50% of FPE and FPIAP patients [12,17] (Table 1). Likewise, a family history of atopy is present in as much as 60% and 80% of first-degree relatives in FPIAP and FPIES, respectively [21,24,26,27].

Interestingly, both FPIES and FPE have been reported in patients with Down syndrome, who may present with a protracted course [15,28,29]. This may be due to intrinsic immune defects seen in these children [30], which include the dysregulated secretion of TNFα and IL-10 [31,32], two key cytokines implicated in FPIES, as described below.

## 4. Pathophysiology

The pathogenesis of non-IgE-GI-FA is poorly understood, but differs from IgE-FA, in that cellular immunity is thought to be driving the allergic inflammatory response while circulating sIgE are notably absent, although a localized IgE-response to the gut has been described and may be a contributing factor [33].

In FPIES, many groups have shown the presence of specific T cell responses to causative antigens [34,35,36,37,38,39,40,41], with an imbalance between exaggerated TNFα secretion by food-specific T cells, and deficient TGFβ responses being consistently reported [36,39,42,43,44,45,46]. This led to an initial understanding of disease as being the result of local T cell infiltration and inflammation causing an increase in intestinal permeability, with a subsequent fluid shift and increase in antigen influx [39,47,48]. However, this simplistic model has recently been challenged in light of conflicting evidence and a lack of specificity of the findings implicating cellular immunity. Th2 responses, which are typical of IgE-FA, have also been described in FPIES, with increased production of IL-4, IL-5, IL-9, and IL-13 cytokines, a phenomenon which may be explained by the high rates of co-atopy seen in these patients [33,36,37,43,49,50]. Conversely, regulatory T cells (Tregs) and IL-10 secretion may play a role in the acquisition of tolerance [37,49,51]. Taken together, these data support the presence of food-specific T cells in FPIES, although more studies are needed to confirm their distinctive role with regards to disease manifestations. Interestingly, other arms of the immune system have recently been studied, and may also contribute to disease. Recent studies using unbiased approaches, such as CyTOF mass cytometry and transcriptional profiling, revealed the broad, systemic activation of the innate immune system after positive FPIES oral food challenges (OFC) [38,52], with the activation of monocytes, neutrophils, eosinophils, natural killer cells, and eosinophils. In another study, higher levels of IL-8 were present in FPIES patients who reacted to an OFC, suggesting neutrophil involvement in those with active disease [37]. The in vitro production of IL-9, a cytokine implicated in the recruitment of mast cells and intestinal anaphylaxis, was also increased in patients with FPIES compared to those with IgE-FA. Baseline tryptase levels were significantly higher in those patients with positive OFC, further supporting the role of mast cells in FPIES [37]. How these innate cells recognize specific food antigens remains, however, enigmatic. Regarding humoral immunity, low levels of neutralizing specific-IgG4 antibodies have been found [37,41,42], while analyses of specific-IgA levels have yielded contradictory results [41,42,53].

In FPE, the structural damage to the jejunal mucosa appears to result from food-specific (mostly cow’s milk) T cell infiltration, causing malabsorption [16,54]. There is strong evidence to support the predominant role of cytotoxic CD8+ T cells in mediating disease [55,56], although an increased density of intraepithelial γδ-TCR+ cells has also been reported [16].

FPIAP is characterized by the dense eosinophilic infiltration of the rectosigmoid mucosa [57,58]. It is still unknown why the inflammatory response is localized to the distal colon in FPIAP, but the high prevalence of this disease amongst breastfed infants suggests that immunologic components found in breastmilk may play a role. One hypothesis is that immunoglobulins secreted in breastmilk may bind dietary proteins, which are only released after being cleaved off by colonic enzymes. Furthermore, the use of antacids, which has been suggested to increase the risk of FPIAP [12], may also enhance the allergenic potential of proteins by preventing their digestion [59]. However, these data need to be confirmed by further studies.

## 5. Clinical Manifestations

While non-IgE-GI-FA comprise distinct clinical entities, they may present with overlapping clinical features. However, there is an apparent gradient of symptom magnitude, with FPIAP and FPIES being at opposing ends of the severity spectrum (Figure 2). Furthermore, some distinctive features (highlighted in Table 1) may help discriminate between the three entities.

In acute FPIES, the hallmark feature is profuse and repetitive vomiting (>95% of patients) occurring 1 to 4 h after food ingestion. Diarrhea can also follow 5 to 10 h later, although this is a less common feature (25–50%). Infants often appear septic, presenting with symptoms of lethargy (65–100%), pallor (30–90%) and hypothermia (5%) [24,26,60,61,62,63,64,65,66,67]. Symptoms may be quite severe, with up to 15% experiencing hemodynamic instability [65,68]. Chronic FPIES, on the other hand, presents with chronic watery diarrhea (occasionally with blood or mucus), intermittent emesis, abdominal distension, and/or poor weight gain. In a subgroup of patients, symptoms progressively worsen and can lead to dehydration (15–45%) and metabolic disturbances (5%) [26,62,64]. Typically, chronic FPIES will occur with persistent exposure to cow’s milk or soy-based formula, while intermittent exposure to solid foods such as rice is more likely to cause an acute presentation [24]. A defining feature of chronic FPIES is the recurrence of symptoms presenting acutely when the trigger food is reintroduced after a period of withdrawal (acute-on-chronic phenotype) [24,69]. Rarely, FPIES has been reported in exclusively breastfed infants [70,71], with cases-series reporting an occurrence of this phenomenon in 0–5% of children [21,26,72]. The age of onset can vary, with FPIES to cow’s milk/soy usually presenting earlier within the first weeks or months of life, while FPIES to solids presents somewhat later, at about 4–7 months of life, probably linked to the timing of the introduction of complementary foods into the diet. Although FPIES generally occurs in early infancy, adult-onset FPIES is now also being increasingly recognized, most frequently triggered by seafood [73,74,75,76]. Finally, a rare occurrence of symptomatic fetal FPIES has recently been reported [77], as well as neonatal FPIES due to intrauterine sensitization [78].

In FPE, patients present with a syndrome resembling that of celiac disease. In most cases, symptoms develop in infants shortly after the introduction of cow’s milk in the diet, with chronic diarrhea and features of malabsorption such as steatorrhea and failure to thrive (FTT), which are responsive to cow’s milk elimination. Vomiting is also frequently reported [15,16,28,79,80,81]. Notably, other extra-digestive symptoms seen in celiac disease, such as dermatitis herpetiformis, are absent. Because of the significant mucosal damage, patients with FPE can also suffer from secondary carbohydrate intolerance [82].

In contrast to FPIES and FPE, FPIAP most often occurs in exclusively breasted infants within the first weeks of life because of indirect exposure to maternal dietary protein via breastmilk, although direct feeding can also trigger symptoms [83]. These infants present with bloody, loose stools, sometimes with mucus. Affected children are however well-appearing, have no severe symptoms of emesis and diarrhea and no significant growth failure [27,58,84,85,86,87,88]. A subset of patients can also present with gagging, food refusal and irritability [12], suggesting that perhaps the allergic inflammation is not entirely restricted to the distal colon. Although FPIAP is most often seen in young infants, it has also been reported to occur in older children, representing 18% of the children evaluated for rectal bleeding in one study [89].

## 6. Reported Food Triggers

Cow’s milk is the most commonly reported food allergy in children [2], of which non-IgE-GI-FA account for up to 40–50% of reactions [90,91,92]. Therefore, unsurprisingly, cow’s milk is responsible for the majority of reactions in FPIES, FPE and FPIAP. However, many other foods have been implicated, as listed in Table 2. In addition to cow’s milk, soy has been frequently reported as a FPIES trigger in countries in which soy formula use is common [93,94], and co-allergy to both foods is seen in up to 40–60% of cases [21,26,95,96]. About a third of patients with cow’s milk/soy FPIES will also react to one or more solid foods [69]. While any solid food can cause FPIES, some overrepresented triggers include grains (rice, oats), vegetables (sweet potato, squash), fruits (banana, avocado), poultry and eggs. Rice is the most commonly reported solid trigger in the United States and Australia, and may be associated with a more severe phenotype [63,97,98,99]. Furthermore, other geographical variations are evident, such as the high prevalence of soy-induced FPIES in the United States [23,25,26,65,95], and of fish and shellfish reactions in Italy and Spain [20,40,60,76,100,101]. These are likely influenced by local dietary practices and may be the result of differences in genetic background [102]. As a whole, common triggers reflect those foods introduced during the first year of life, as index reactions are unlikely to occur in foods introduced thereafter [26,61,93]. While the majority (65–80%) of children react to a single food, most commonly cow’s milk, about 10% of children will react to three foods or more. In contrast, a majority (up to 80%) of those with FPIES to solid food tend to have multiple food triggers [24,26,65]. Frequently, children will react upon their first exposure to the incriminated food [64], indicating that prior sensitization is either not required or occurs through other routes such as the skin.

While there is a paucity of recent studies evaluating FPE food triggers, older studies have consistently reported cow’s milk to be the main culprit [15,18,28,103]. In a case-series of 54 infants with cow’s milk FPE, co-allergy to soy was reported in 4/35 (11%) of those tested, and to wheat in 7/19 (37%) [28]. Other reported triggers included eggs (*n* = 2), bananas (*n* = 2), and meat (*n* = 1).

Finally, FPIAP is most frequently caused by indirect exposure to cow’s milk (and other foods) via breastmilk, occurring in exclusively breasted infants in over one-half of cases [84]. Somewhat less commonly, FPIAP can result from direct ingestion of cow’s milk (44%) or soy-based formula (7%) [83]. Other culprit foods include soy, egg, wheat and corn [12,84,85,89].

## 7. Diagnosis

The diagnosis of non-IgE-GI-FA remains, for the most part, a clinical one, with the exception of FPE, in which histological confirmation is usually required. Other etiologies presenting with a similar clinical picture should also be excluded. Optimal diagnosis and management may require the expertise of a multidisciplinary team (Figure 4).

The diagnosis of FPIES is established with the presence of a constellation of symptoms concordant with FPIES, and the resolution of symptoms upon the removal of offending foods from the diet. In an effort to standardize the diagnosis of acute FPIES in light of latest available data, recent international consensus guidelines based on expert opinion have defined major and minor criteria (Table 3) [68], although the accuracy of these diagnostic criteria has not yet been prospectively validated. While the OFC is no longer mandatory for diagnosis confirmation based on these criteria, it should be strongly considered when only a single episode has occurred, or when the causative food remains elusive. Tentative diagnostic criteria have also been proposed for chronic FPIES, with pathognomonic features being the rapid resolution of symptoms (within days) after the withdrawal of offending foods, and the acute presentation when the food is later reintroduced after a period of elimination [68]. In contrast to acute FPIES, the OFC is mandatory for chronic FPIES diagnosis, which is intended to decrease the frequent overdiagnosis found with this entity.

There are no accepted diagnostic criteria available for FPE and FPIAP, although some elements routinely used in clinical practice to help support the diagnosis of these diseases are listed in Table 3. FPE presents in young infants (<9 months) with typical symptoms of vomiting and intestinal malabsorption. In addition, a biopsy suggestive of FPE helps to confirm the diagnosis [81]. In FPIAP, the causative association between the rectal bleeding and antigenic food must be established with a resolution of symptoms upon the elimination of the offending foods, and, preferably, with the documented recurrence of symptoms when foods are re-introduced [106]. Furthermore, other causes of hematochezia, such as anal fissures, must be excluded.

Diagnosis of non-IgE-GI-FA may be delayed, because foods such as grains, vegetables and poultry are often perceived as being hypoallergenic, a misconception based on their infrequency as IgE-FA triggers, or because the reactions do not occur immediately after food ingestion, thus making the association more difficult. Furthermore, the lack of awareness of these entities by health care providers may also be a contributing factor [4,5].

## 8. Oral Food Challenge

OFCs represent the gold standard to confirm the diagnosis of FPIES, and are particularly important to avoid misdiagnosis with other common gastrointestinal diseases of infancy (Table 4). Of importance, OFC performed for suspected FPIES is considered a high-risk procedure. The consensus protocol is to administer 0.3 g/kg (0.06–0.6 g/kg) of food protein (max: 3 g of protein, 10 g of total food or 100 mL of liquid) in three equal doses over 30 min and observe the patient for 4–6 h for potential reactions [68]. In patients with severe reactions, a lower starting dose and a longer observation period may be used. Some centers have opted for performing a 1-dose OFC protocol instead, with success [93]. However, a graded OFC should always be performed in those patients with the presence of food sIgE, because of the risk of an immediate reaction [107].

While OFC protocols may differ slightly between centers, an overriding principle is that OFC should be administered by trained professionals in a facility equipped with intravenous (IV) fluid resuscitation material, as this intervention may be required in up to 50% of patients [68]. Whether a peripheral IV line should systematically be secured prior to undertaking the OFC remains controversial, but should be considered in patients in whom a difficult IV access is anticipated, and in those with a history of severe reactions. Nonetheless, the severity of the index reaction may not reliably predict the severity of the OFC reaction [93]. If symptoms are mild, PO fluids may be sufficient [19]. Ondansetron, a serotonin receptor antagonist used as a potent anti-emetic drug in patients receiving chemotherapy [108], may be helpful in mitigating acute symptoms when used as an adjunctive therapy [109,110]. Ondansetron should not be given to children under 6 months of age due to the lack of safety data, and should be administered with caution in predisposed patients due to the potential for QT interval prolongation. While methylprednisone has been used in an attempt to suppress the presumed cell-mediated inflammation, there are no studies to support its use. Furthermore, the current evidence does not support a role for epinephrine autoinjectors and antihistamines in the management of acute FPIES [68].

Supervised OFC is generally not required for the diagnosis of FPE and FPIAP [111]. However, in FPIAP, an early home challenge has been proposed to confirm the diagnosis after a short period of food elimination, to minimize the risk of misdiagnosis [13,112].

## 9. Paraclinical Investigations and Biomarkers

### 9.1. Laboratory Findings

Laboratory studies are somewhat helpful in supporting the diagnosis of non-IgE-GI-FA, although the findings are usually nonspecific. In FPIES, blood testing can uncover anemia, hypoalbuminemia, thrombocytosis, and a high white blood cell count with left shift [63,68]. With increased severity, metabolic acidosis and methemoglobinemia may also develop [113]. In FPE, iron-deficiency, anemia and hypoproteinemia are frequently present [16,114], while these are uncommon in FPIAP (<15%) [27,85,115,116]. Peripheral eosinophilia can be found in FPIES, FPE and FPIAP, although it is a more prominent feature in the two latter entities [28,58,63,83,85,117]. Lymphocyte transformation tests (LTT) remain experimental, and as such, are not recommended [118].

### 9.2. Allergy Testing

Allergy testing, which includes epicutaneous skin testing and the detection of serum food-specific IgE, is usually negative, and not warranted for the investigation of non-IgE-GI-FA [2,3,8]. However, up to 25% of FPIES patients will develop sIgE to their food trigger [23,26,64]. These patients have what is termed an atypical form of FPIES, which can present with a protracted course and may increase the risk of developing immediate allergic reactions to the culprit food [26,95]. Therefore, although allergy testing is usually not recommended at the first evaluation of FPIES, it can be performed during follow-up to exclude atypical FPIES, particularly before an OFC to adapt the protocol if need be [68]. Atopy patch testing (APT) has shown conflicting efficacy in the diagnosis of FPIES, with one study concluding that APT reliably predicted 85% of OFC outcomes [119], while two other studies found no such benefit [95,120]. These tests are therefore currently not recommended.

### 9.3. Stool Studies

When conducted in FPIES patients, stool evaluation may reveal nonspecific findings such as neutrophils, eosinophils, Charcot–Leyden crystals, and reducing substances [68]. Several studies have used occult blood testing to aid in the diagnosis of FPIAP [12,121], although occult blood may also be found in FPIES and other diseases. Likewise, while it is not specific to these diseases, fecal calprotectin was found to be elevated in both FPIES and FPIAP, indicative of gut mucosal inflammation [122,123,124,125]. However, this test should not be used before one year of age, due to the wide variability of values reported in infants, and the absence of validated normal ranges [126,127].

### 9.4. Radiologic Evaluation

Radiologic evaluation is not part of the usual disease workup, unless used to rule-out other disease processes. If abdominal X-rays are performed in children with FPIES, these may reveal nonspecific findings such as air-fluid levels, colon narrowing and thumbprinting, and duodenojejunal thickening of the plicae circulares [128]. Furthermore, intramural gas has also been reported, and may lead to a misdiagnosis of necrotizing enterocolitis (NEC) [129,130]. Recent ultrasonographic studies of 16 patients with non-IgE-GI-FA found that 100% of patients exhibited small intestinal wall thickening and poor peristalsis, which disappeared after culprit food elimination [131]. However, more studies are needed to corroborate the use of intestinal ultrasound before it can be used routinely.

### 9.5. Endoscopic Evaluation

Endoscopic evaluation is most important when FPE is suspected, as it supports the diagnosis. Histology findings typically phenocopy those seen in celiac disease, although they tend to be less severe [132]. These include varying degrees of jejunal villous atrophy and crypt hyperplasia [28,82], with the villus-to-crypt ratio being a sensitive marker of morphologic change due to jejunal damage [133]. In FPIAP, rectosigmoidoscopy may reveal mucosal congestion with petechial areas [134]. The inflammation is characterized by eosinophilic infiltration and lymphonodular hyperplasia [84,89,134]. The inflammatory cell infiltrate seen in FPIES includes lymphocytes, plasma cells, eosinophils, and mast cells [68].

### 9.6. Other Biomarkers

Interestingly, C-reactive protein (CRP), an acute phase reactant produced by the liver in response to the pro-inflammatory cytokine IL-6, has been reported to be marginally elevated following positive OFC in patients with FPIES [72,135,136,137]. However, while low elevation can be found, a high CRP (>20 mg/dL) should prompt one towards an alternative diagnosis [63]. Platelets, another acute phase reactant, are often elevated in FPIES, with one study reporting a significantly lower mean platelet volume (MVP) than in bacterial sepsis, which may help discriminate between both conditions [63].

## 10. Differential Diagnosis

The differential diagnoses of non-IgE-GI-FA are listed in Table 4. Broadly, entities that may mimic acute FPIES include those with acute vomiting in an ill-appearing child, while those resembling chronic FPIES and FPE comprise a variety of diseases leading to chronic GI symptoms with FTT. Alternative sources of hematochezia in infants can also simulate FPIAP, and should be excluded. Infectious causes such as viral and bacterial gastroenteritis are frequent in this age group, and can present with similar features as FPIES. In a moribund infant, the diagnostic workup should include sepsis evaluation. Helpful distinguishing features of acute FPIES include the timing after food ingestion, a history of repeated episodes, and the rapid resolution of symptoms within 24 h, which contrasts with the prolonged symptoms usually seen in infectious causes. Fever is not a common feature of FPIES, although it has been described in Japanese and Chinese studies, a phenotype that appears to be specific to these regions [72,135,138,139]. Monogenetic diseases, such as inborn errors of metabolism, primary immunodeficiencies, and cystic fibrosis, can also present with similar sepsis-like features and failure to thrive, and may not always be detected by newborn screening [140]. Mechanical obstruction caused by pyloric stenosis, intussusception, volvulus or Hirschsprung disease [141] may also present with acute and chronic vomiting, although in contrast to non-IgE-GI-FA, the symptoms will not be food-specific. Finally, IgE-FA, which can be life-threatening, is not always easily discernible from FPIES. However, the immediate occurrence of symptoms (within an hour of food ingestion), and other systemic features such as hives, angioedema, and respiratory symptoms, are uniquely present in IgE-FA.

## 11. Natural History

The natural history of non-IgE-GI-FA is mostly favorable, with the vast majority being outgrown by school age.

FPIES generally resolves before 3 to 5 years of age, but this may vary by type of food (i.e., solid vs. liquid) and geographical location [61]. An older age at diagnosis and atypical FPIES are associated with a prolonged course [26,75]. Furthermore, those with atypical FPIES are at increased risk of shifting from a non-IgE-mediated disease to an immediate, IgE-mediated phenotype [19,23,62,142,143]. The development of circulating sIgE in these patients with subsequent IgE-FA is perplexing, and may be indicative of some level of fluidity between the pathogenesis of both types of allergy. In line with this hypothesis, some cases of IgE-FA conversely shifting to a non-IgE-FA phenotype have also been described [144,145,146]. An alternative explanation is that the avoidance of the allergenic protein during a window of susceptibility in infancy may in fact increase the risk of developing IgE-FA in susceptible individuals [147,148,149].

In contrast to the permanent course seen in celiac disease, FPE is usually transitory and typically resolves by 1–2 years of age [28,79,150]. However, the disease may persist later into childhood in some cases [151,152,153].

Similarly, FPIAP rarely persists beyond 1–2 years of age [83]. In one large series of children with FPIAP, the mean age at which cow’s milk was successfully reintroduced was 11 months [12]. Notably, in 23 patients (15%), in whom the diet was not restricted at all, the symptoms eventually subsided in all patients, who subsequently tolerated cow’s milk throughout infancy, suggesting a benign course, despite continued exposure in a subset of patients [12]. However, recent studies suggest that patients with FPIAP may be at an increased risk of childhood functional GI disorder, especially in those with a more severe and protracted initial disease [154].

## 12. Management

### 12.1. Food Elimination

The cornerstone of the management of non-IgE-GI-FA is the removal of offending foods from the diet. Depending on the severity of the symptoms and on the quantity of triggering foods, two different strategies can be adopted. A “bottom-up approach”, in which causal foods are successively eliminated without broad restrictions of unsuspected triggers, is used in most cases. Indeed, one general principle of the management of food allergies is that the avoidance of causative foods only, without any restriction of foods that are tolerated, should be the norm. However, a “top-down approach” may be warranted in most severe cases where FTT and dehydration are prominent. This approach consists of an initial avoidance of a wide variety of foods, sometimes starting with an elemental diet, with the sequential reintroduction of individual foods, while carefully monitoring for recurrence of symptoms [10,155]. Counselling with a nutritionist is recommended because of the high risk for nutritional deficiencies (discussed below).

Cow’s milk is the most frequent trigger in FPIES, FPE and FPIAP, and should be replaced with an extensively hydrolyzed formula (EHF). While most patients are responsive to EHF, 10–20% of patients may require an amino acid-based formula (AAF) (Table 5) [12,21,24,68,156,157]. Of note, a change in stool consistency is frequent with hypoallergenic formulas, and is not indicative of ongoing colitis. In all cases, partially hydrolyzed formulas should be avoided because of their residual antigenic content, with a concentration of intact cow’s milk protein 1000–100,000-fold greater than that of EHF [158]. When tolerated, a soy-based formula is an acceptable alternative, although it is usually avoided because of the high co-reactivity with cow’s milk, seen in up to 40–60% of patients with FPIES, and in 10–30% of patients with FPE and FPIAP [96,158,159,160]. However, co-sensitization to cow’s milk and soy has mainly been reported in studies from the United States, and may be less frequent in other populations [19,62,104,105].

In contrast to FPIAP, symptomatic FPIES is very rare in the breastfed infant. Therefore, the mother should not avoid trigger foods unless symptoms are clearly documented [68,106]. Although no threshold dose has reliably been established in FPIES, only 0.15 g of protein/kg was sufficient to trigger 13/13 acute FPIES reactions to OFC in one study [119]. A strict avoidance of all offending foods should therefore be adopted. While avoidance of foods with precautionary allergen labeling (“may contain traces of”) is generally not necessary [68], it is usually recommended that extensively baked and processed foods be avoided in FPIES. Nevertheless, studies have shown that a number of patients with FPIES can tolerate these foods in baked form [161,162,163,164]. The long-term outcome of this practice is yet unknown, and OFC should be performed with caution, because some children may still react. Acute FPIES reactions should be treated as per positive OFC (see above). Epinephrine autoinjectors should only be prescribed if there is a concomitant IgE-mediated food allergy. The education of caregivers is central to a comprehensive management plan, and a written emergency plan of action should be given, as well as an explicative letter of the diagnosis and management plan to present to emergency room (ER) physicians who may not be familiar with FPIES. As per IgE-FA, a medical ID bracelet can be useful.

In FPIAP, the vast majority of exclusively breastfed infants will respond to the maternal elimination of all milk products (including butter) [84,165]. Occasionally, the elimination of multiple foods may be required, usually soy and/or egg [84]. Recent European guidelines recommend a 2–4 weeks maternal elimination diet, followed by an attempt to re-introduce foods in order to confirm the diagnosis [106]. In those in whom the maternal restriction diet is unsuccessful or too cumbersome, infants can be placed on an EHF instead. Because episodes of rectal bleeding in infants are often self-resolving, some authors have instead suggested using a “watch and wait” approach during the first month of symptoms, before undertaking an elimination diet [112]. In those with a symptoms duration that is greater than a month, hemoglobin levels should be assessed before undertaking a short cow’s milk elimination trial.

### 12.2. Nutritional Impact

It is essential to accurately diagnose non-IgE-GI-FA and to offer proper nutritional guidance, as both continued exposure and dietary manipulation can have detrimental effects on health. Efforts should therefore be put in place to limit unnecessary elimination diets, as even single-food avoidance can lead to significant nutritional deficiency [166], although those who avoid a greater number of foods may be at an increased risk [167]. Children with non-IgE-GI-FA can experience poor growth, which may be attributed to increased losses due to intestinal malabsorption and/or to limited food intake due to feeding difficulties, which are found in up to 30% of these patients [64,168]. Consequently, both weight [64,168,169] and height [169] may be affected. Deficient intake of indispensable micronutrients, including vitamin D, calcium, zinc, selenium [170,171,172,173], has also been reported, negatively impacting bone health [173].

### 12.3. Culprit Food Reintroduction

In the event of a positive OFC in FPIES, follow-up challenges should be performed, to determine whether FPIES has resolved. While there is no international consensus regarding the best time to perform an OFC, a repeat OFC every 12–18 months after the latest reaction is generally proposed in the United States, although this may vary depending on the type of food, the severity of the initial reaction, and distinct geographical characteristics [68].

In FPIAP, cow’s milk can be progressively re-introduced in a stepwise manner by 12 months of age in most patients [84,106,174,175]. If the diagnosis is unclear, re-introduction can be attempted earlier, because some infants with mild rectal bleeding may have a self-limiting course [11,13]. In patients with initially mild symptoms of FPE or FPIAP without any features suggestive of IgE-FA or FPIES, reintroduction can usually be done at home. In the case of failure, the trigger should be removed once again from the diet, with new attempts at reintroduction every six months.

### 12.4. Introduction of Weaning Foods

While there is no clear guidance regarding the ideal sequence of the introduction of complementary foods in patients with FPIES, current guidelines advocate in favor of introducing low-risk foods first, such as vegetables (broccoli, cauliflower) and fruits (blueberries, strawberries) [68]. Moderate and high-risk foods may then be serially introduced over the following months, when developmentally appropriate. Importantly, foods with different colors, flavors and textures should continuously be offered, in order to minimize the risk of aversive feeding behaviors. Furthermore, the introduction of iron-rich foods should also be encouraged, because the onset of FPIES often correlates with hemoglobin nadir, leaving patients vulnerable to anemia [176].

In FPE and FPIAP, food diversification can follow usual recommendations without any particular restrictions.

## 13. Quality of Life

While the impact of IgE-mediated food allergies on quality of life (QoL) has long been established, the effect of non-IgE-GI-FA has only been recently studied. Strikingly, one cross-sectional study of 52 children with non-IgE-GI-FA reported lower physical QoL scores compared to children with IgE-FA [177]. Similarly, another prospective observational study of 123 children with non-IgE-GI-FA revealed inferior QoL scores in all evaluated domains compared to children with sickle cell anemia, including spheres of physical functioning, emotional functioning, and worry [178]. Furthermore, QoL scores in those domains were also lower than in patients with intestinal failure. Unsurprisingly, a greater number of foods excluded was associated with lower QoL in both studies.

## 14. Role of Food Allergens in Other Common Pediatric Gastrointestinal Disorders

Food allergens may play a role in a subset of children with common gastrointestinal disorders, such as gastroesophageal reflux disease (GERD), irritable bowel syndrome (IBS), constipation and colic, although further studies are required to better elucidate the underlying pathomechanisms involved and whether these are immunologic and/or non-immunologic adverse reactions [10,179].

A subset of infants with GERD may have an underlying cow’s milk allergy, especially in patients with severe presentations, atopic dermatitis and FTT [179,180,181]. Cow’s milk feeding can result in gastric dysrhythmia, delayed gastric emptying, and increased episodes of reflux in milk-allergic infants [179,182,183]. However, there is no strong evidence supporting immunological mechanisms for GERD triggered by cow’s milk; the efficacy of hydrolysates in reducing regurgitations in infants with GERD may be due to enhanced gastric emptying compared to native cow’s milk protein [10,184].

IBS is a multifactorial gastrointestinal disorder which involves digestive symptoms that can be triggered by different foods, such as fermentable oligosaccharides, disaccharides, monosaccharides and polyols (FODMAPS) [37]. IBS can overlap with non-celiac gluten sensitivity, with controlled studies showing an improvement of patients after the dietary exclusion of wheat [37]. Interestingly, children with atopic disease are at greater risk of developing IBS than non-atopic children [185]. A subset of patients with IBS have positive colonoscopic food allergen provocation with localized mucosal wheal and flare reactions [186] and the improvement of symptoms with cromoglycates [187], suggesting a potential role of food allergens in the pathogenesis of IBS. A recent study found the immediate disruption of the small intestinal barrier by confocal laser endomicroscopy after oral provocation with common allergens (wheat, milk, egg, soy) in skin test/specific IgE negative patients with IBS [188]. Further larger studies and randomized controlled trials are needed to better investigate the exact role and pathophysiology of food allergens as triggers in IBS patients.

The role of food allergens as a cause of constipation is controversial. Transition from breast milk to cow’s milk formula can be associated with mild constipation and increased stool hardness [189]. Prospective trials have found that cow’s milk free diets could improve constipation in a proportion of children, and children responding to this diet were more likely to be atopic [37]. A study in children found increased resting anal pressure and a reduced mucous gel layer, which could be attributed to allergic inflammation triggered by milk [190]. However, the potential effect of cow’s milk could be explained by non-immunologic mechanisms leading to changes in stool consistency [37].

Infantile colic is characterized by paroxysmal inconsolable crying during the early weeks of life [37]. The role of food allergens as triggers of colic is controversial, although food triggers may be relevant in a subset of infants with severe colic and atopic risk factors [37,191]. These infants may potentially benefit from a short trial of extensively hydrolyzed or soy formula [191]. However, a recent meta-analysis found sparse evidence for the effectiveness of dietary modifications for the treatment of infantile colic due to confounding factors (most cases of infantile colic spontaneously improve within a short time frame without elimination diets) [192]. Future larger studies are needed to better define the role of food allergens in infantile colic.

## 15. Future Perspectives

While tremendous advances have been made in our understanding of non-IgE-GI-FA in the past decade, many questions remain unanswered. For one, the pathogenesis of non-IgE-GI-FA remains poorly elucidated. Multiple factors have hindered the ability to properly study these diseases, including (i) their relatively low prevalence (ii) their occurrence in young infants, where access to blood cells is challenging, (iii) a disease process potentially localized to the GI tract, which requires invasive procedures for tissue sampling and (iv) a lack experimental animal models currently available. Nonetheless, a better understanding of the immunological components driving these diseases is a high priority, since it may help uncover noninvasive biomarkers for efficient diagnosis, which are currently lacking, and provide guidance towards future preventive and therapeutic opportunities.

As in most acquired diseases, the ideal goal in food allergy is prevention. Recent, ground-breaking research has demonstrated that the early introduction of allergenic foods such as peanuts [149] and eggs [193] may reduce the risk of subsequently developing IgE-FA. However, no such preventive measures have been found effective so far in FPIES [19,24]. In FPIAP, the concurrent use of formula while breastfeeding may be protective [12], although more studies are needed to confirm this.

If prevention cannot be achieved, another objective may be to accelerate the acquisition of oral tolerance. Preliminary studies indicate that probiotic supplementation may accelerate the resolution of non-IgE-GI-FA to cow’s milk [124,194,195]. While the introduction of baked egg and milk can facilitate the resolution of IgE-FA [196], it is unknown whether this phenomenon is also present in non-IgE-GI-FA, and if early exposition to baked forms could help prevent a shift to IgE-FA. Interestingly, the early introduction of solid foods (<5.5 months) may accelerate the resolution of FPIAP [85]. Oral immunotherapy (OIT) has not yet been attempted in non-IgE-GI-FA, potentially because spontaneous tolerance is achieved relatively early in most cases, although it may be an interesting therapeutic avenue to study in forms that persist beyond infancy. Finally, the blockade of cytokines implicated in FPIES such as TNFα, IL-1β and IL-6 are exciting potential therapeutic targets [52].

## 16. Conclusions

In summary, non-IgE-GI-FA are frequently implicated in causing gastrointestinal symptoms in children, and are likely underdiagnosed. Owing to a paucity of available literature, evidence-based protocols to diagnose and treat these diseases are lacking. Currently, diagnosis is made clinically, although diagnostic criteria are evolving to better capture the various phenotypes as our understanding of these diseases progresses. The management relies upon avoidance of dietary triggers, and requires a multidisciplinary approach. Future studies are needed to better understand pathogenesis, to determine additional biomarkers for better diagnosis, and to optimize management practices. Finally, food allergens may also be implicated in a subset of patients with common GI disorders, although the current evidence to support an underlying immunologic pathomechanism is limited.

## Figures and Tables

**Figure 1 nutrients-12-02086-f001:**
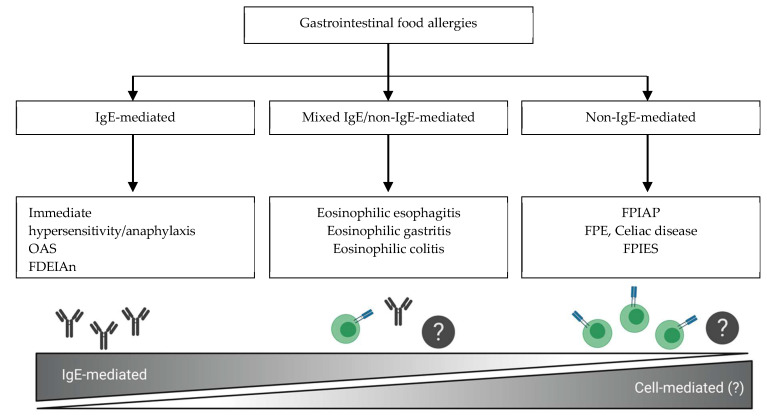
Classification of gastrointestinal food allergies. FDEIAn, food-dependent exercise-induced anaphylaxis; FPE, food protein-induced enteropathy; FPIAP, food protein-induced allergic proctocolitis; FPIES, protein-induced enterocolitis syndrome; IgE, immunoglobulin E; OAS, oral allergy syndrome.

**Figure 2 nutrients-12-02086-f002:**
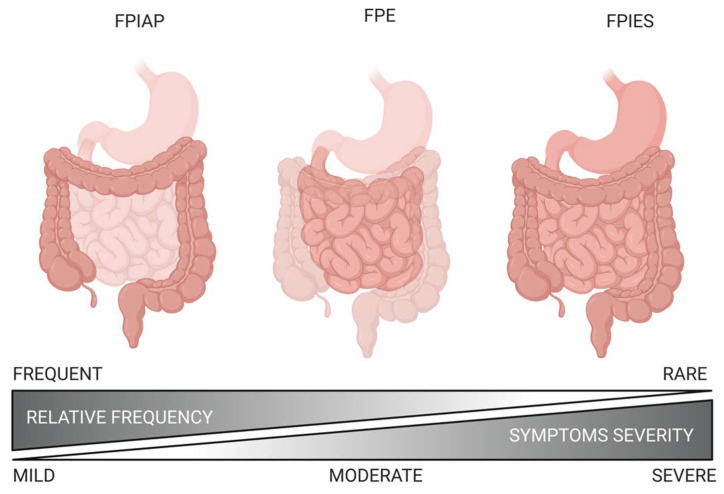
Gastrointestinal organs affected in the different non-IgE-mediated gastrointestinal food allergies. FPIAP and FPE affect the colon and small intestine, respectively, while in FPIES, the whole gastrointestinal tract can be affected. FPE, food protein-induced enteropathy; FPIAP, food protein-induced allergic proctocolitis; FPIES, protein-induced enterocolitis syndrome.

**Figure 3 nutrients-12-02086-f003:**
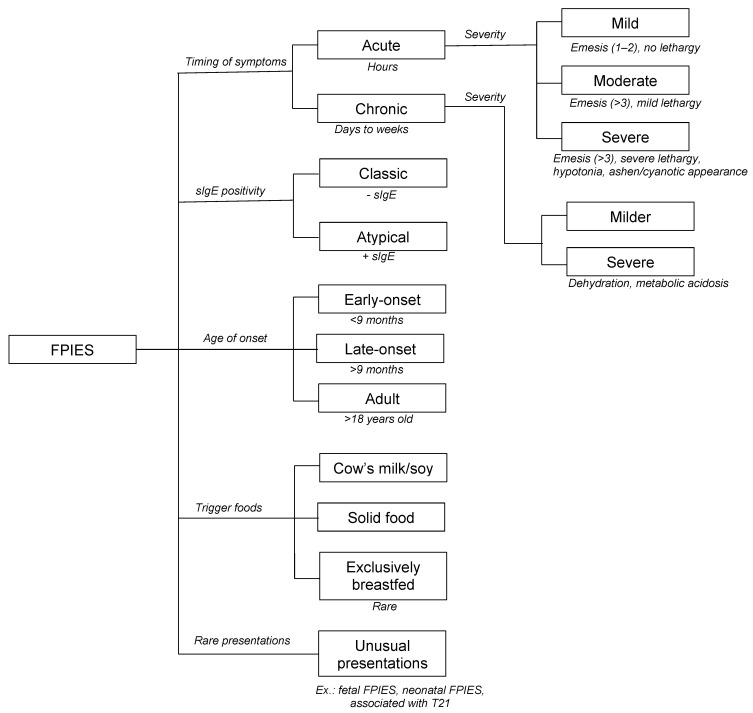
Classification scheme of FPIES. FPIES, food protein-induced enterocolitis syndrome; sIgE, food-specific immunoglobulin E; T21, trisomy 21 (Down syndrome).

**Figure 4 nutrients-12-02086-f004:**
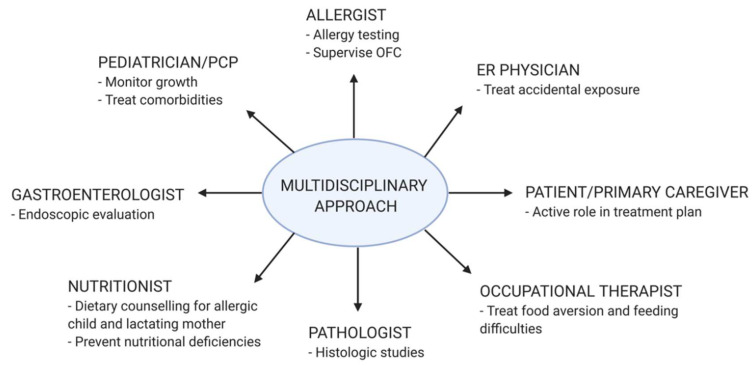
Multidisciplinary approach for diagnosis and management of non-IgE-mediated gastrointestinal food allergies. ER, emergency room; OFC, oral food challenge; PCP, primary care physician.

**Table 1 nutrients-12-02086-t001:** Clinical and laboratory features of non-IgE-mediated gastrointestinal food allergies.

.	FPIES	FPE	FPIAP
Age of presentation	Cow’s milk/soy: First weeks-months of life Solids: 4–7 months Can also occur in adults	2–24 months Can also occur in older children	First weeks-months of life (<6 months) Can also occur in older children
Top culprit foods	Cow’s milk, soy (C > A) Rice, poultry, fish, fruits, vegetables (A > C)	Cow’s milk, soy Wheat, egg	Cow’s milk, soy Egg, corn, wheat
Multiple foods	Frequent ≥3 foods: 5–10%	Rare	Occasional
Feeding at onset	Formula	Formula	Exclusively BF (>50%)
Clinical presentation	(A): repeated vomiting, diarrhea, dehydration (shock: 15%), lethargy, pallor, hypothermia (C): intermittent vomiting, diarrhea, FTT	Diarrhea, intermittent vomiting, FTT, malabsorption (steatorrhea), bloody stools (rare)	Blood/mucus streaked stools, mild diarrhea Otherwise well-appearing
Co-morbid atopy	40–60% Familial: 40–80%	20–40%	25–50% Familial: 30–60%
Laboratory anomalies	Anemia (C) Eosinophilia (C) Neutrophilia (A, C) Thrombocytosis (A) Methemoglobinemia (A, C) Metabolic acidosis (A, C)	Anemia Hypoalbuminemia Iron deficiency	Mild anemia Hypoalbuminemia (rare) Eosinophilia
Stool studies	Occult blood (A, C) PMN (A, C) Eosinophils (A, C) Reducing substances (C)	Fecal fat Low d-Xylose excretion	Gross/occult blood Eosinophils
Endoscopy/Histology	Friable mucosa Ulceration Villous atrophy Crypt abscesses Inflammatory cell infiltrates	Villous atrophy Crypt hyperplasia Lymphocytic infiltrate	Mild, focal colitis Eosinophilic infiltration Lymphonodular hyperplasia
Allergy evaluation	Negative; sIgE+ in 25%	Negative (not recommended)	Negative (not recommended)
Diagnosis	Clinical +/− OFC	Clinical & histological	Clinical +/− OFC
Treatment	Avoidance of offending foods	Avoidance of offending foods	Avoidance of offending foods (maternal exclusion diet if BF)
Time to improvement	(A) 4–12 h (<24 h) (C) 3–10 days	Several weeks (1–2 weeks)	72 h (up to 2 weeks)
Natural history	Resolution < 3–5 y Later if sIgE+ or solid foods	Resolution < 1–2 years	Resolution < 1–2 years

A: acute; APT, atopy patch test; BF, breastfed; C: chronic; FTT, failure to thrive, OFC, oral food challenge; PMN, polymorphonuclear leukocyte; sIgE, specific immunoglobulin E.

**Table 2 nutrients-12-02086-t002:** Foods commonly implicated in non-IgE-mediated gastrointestinal food allergies in select large case-series studies.

	FPIES						FPE	FPIAP	
Country	USA ^1^	UK ^2^	Spain ^3^	Italy ^4^	Australia ^5^	Turkey ^6^	Finland ^7^	USA ^8^	Turkey ^9^
*N* (total)	*N* = 1340	*N* = 54	*N* = 336	*N* = 66	*N* = 265	*N* = 27	*N* = 54	*N* = 95	*N* = 359
	%	%	%	%	%	%	%	%	%
Cow’s milk	19–67	46	26–38	67	20–33	74	100	65	91–100
Soy	8–41	11	0–1	4	5–34	-	11	3 *	0–3 *
Rice	19–53	4	1–10	4	40–45	4	-	-	-
Oat	16–37	6	0–1	-	6–9	-	-	-	-
Wheat	1–16	11	0–1	2	0–3	4	37	-	0–4
Corn	2–8	2	0–3	2	0–1	-	-	6	-
Eggs	0–23	13	10–21	6	0–12	-	4	18	7–22
Fish/Shellfish	1–15	15	34–54	12	3–5	15	-	-	0–2
Poultry	5–10	7	1–4	3	3–8	-	-	-	0–3
Meat	3–18	4	1	-	3–4	-	2	-	0–10
Sweet potato	4–22	-	-	-	3–6	-	-	-	-
Potato	2–8	2	0–1	-	0–2	4	-	-	0–2
Squash	0–12	-	-	-	-	-	-	-	-
Carrot	0–7	4	0	-	0–1	-	-	-	0–1
Banana	4–24	6	0–1	3	3–4	4	4	-	-
Avocado	0–16	-	-	-	0–2	-	-	-	-
Apple	0–11	2	0–1	-	0–2	-	-	-	0–1
Pear	0–9	-	0–1	-	0–3	-	-	-	-

* Soy allergy likely underrepresented by these studies. ^1^ From Ruffner et al. [95] (*n* = 462), Caubet et al. [26] (*n* = 160), Blackman et al. [65] (*n* = 74), Maciag et al. [25] (*n* = 441), Su et al. [64] (*n* = 203); ^2^ From Ludman et al. [66] (*n* = 54); ^3^ From Vazquez-Ortiz [101] (*n* = 81), Diaz et al. [60] (*n* = 120), Pérez Ajami et al. [104] (*n* = 135); ^4^ From Miceli Sopo et al. [67] (*n* = 66); ^5^ From Mehr et al. [97] (*n* = 35), Mehr et al. [21] (*n* = 230); ^6^ From Arik Yilmaz et al. [105] (*n* = 27); ^7^ From Kuitunen et al. [28] (*n* = 54); ^8^ From Lake et al. [84] (*n* = 95); ^9^ From Kaya et al. [87] (*n* = 60), Arik Yilmaz et al. [105] (*n* = 37), Erdem et al. [86] (*n* = 77), Cetinkaya et al. [85] (*n* = 185).

**Table 3 nutrients-12-02086-t003:** Diagnostic criteria of non-IgE-mediated gastrointestinal food allergies.

Acute FPIES ^1^	
Major Criteria, PLUS	Minor Criteria (≥3 Occurring with Episode)
1. Vomiting 1–4 h after suspect food ingestion AND 2. Absence IgE-mediated allergic symptoms	1. ≥2 episodes with same food 2. 1 episode with a different food 3. Lethargy 4. Pallor 5. Need for ER visit 6. Need for IV fluid support 7. Diarrhea within 24 h (usually 5–10 h) 8. Hypotension 9. Hypothermia
Chronic FPIES ^2^	
Symptoms and severity	Criteria
Milder (lower doses with intermittent ingestion): 1. Intermittent vomiting and/or diarrhea 2. FTT 3. No dehydration or metabolic acidosis Severe (higher doses with chronic ingestion): 1. Intermittent but progressive vomiting and diarrhea (occasionally with blood) 2. Possible dehydration and metabolic acidosis	1. Resolution of symptoms within days after elimination of offending food(s) 2. Acute recurrence of symptoms (vomiting in 1–4 h, diarrhea in <24 h, usually 5–10 h) when the food is reintroduced 3. Confirmatory OFC, or presumptive diagnosis if OFC not performed
FPE ^3^	
1. Generally <9 months of age at diagnosis, but can also present in older children 2. Repeated exposure to causative foods elicits gastrointestinal symptoms (predominantly vomiting and FTT), without alternative cause 3. Histologic confirmation of the diagnosis in a symptomatic child by biopsy of the small bowel showing villous injury, crypt hyperplasia and inflammation 4. Clinical and histological improvement after removal of offending food(s) 5. Exclusion of alternative causes	
FPIAP ^4^	
1. Mild rectal bleeding in an otherwise healthy infant 2. Resolution of symptoms after elimination of offending food(s) (if exclusively breastfed, resolution after a maternal elimination diet) 3. Recurrence of symptoms upon reintroduction of culprit food(s) in the diet (preferable) 4. Exclusion of other causes of rectal bleeding	

^1^ Major criterion must be met (both) plus at least three minor criteria (adapted from the International consensus guidelines [68]); ^2^ General criteria because of paucity of available data (adapted from the International consensus guidelines [68]); ^3^ There are no defined criteria in the literature for FPE diagnosis. The elements listed here are generally used for diagnosis in clinical practice (adapted from [81]); ^4^ There are no defined criteria in the literature for FPIAP diagnosis. The elements listed here are generally used for diagnosis in clinical practice (adapted from EEACI position paper [106]); ER, emergency room; FTT, failure to thrive; IgE, immunoglobulin E; IV, intravenous; OFC, oral food challenge.

**Table 4 nutrients-12-02086-t004:** Differential diagnosis of non-IgE-mediated gastrointestinal food allergies.

	Acute FPIES	Chronic FPIES	FPE	FPIAP
Allergic	Anaphylaxis Eosinophilic gastroenteropathies	FPIAP FPE Eosinophilic gastroenteropathies	Celiac disease Chronic FPIES Eosinophilic gastroenteropathies	FPIES FPE Eosinophilic gastroenteropathies
Infectious	Sepsis Viral/bacterial/ parasitic gastroenteritis	Viral/bacterial/ parasitic gastroenteritis	Viral/bacterial/ parasitic gastroenteritis	Viral/bacterial/ parasitic gastroenteritis
Gastrointestinal	Hirschsprung Pyloric stenosis Intussusception Volvulus NEC	GERD Hirschsprung Pyloric stenosis VEOIBD Cystic fibrosis	VEOIBD Cystic fibrosis	Anal fissure Swallowed maternal blood NEC Intussusception Volvulus Meckel diverticulum Intestinal duplication kyst Infantile polyp VEOIBD
Metabolic	Inborn errors of metabolism T1DM	Inborn errors of metabolism T1DM	Inborn errors of metabolism Congenital disaccharidase deficiency T1DM	-
Hematologic	Congenital methemoglobinemia	Congenital methemoglobinemia	-	Coagulation defect Thrombocytopenia
Neuro- logic	Cyclic vomiting Intracranial mass	Cyclic vomiting Intracranial mass	-	-
Cardiovascular	Congenital heart defect Cardiomyopathy Arrythmia	Congenital heart defect Cardiomyopathy	-	Vascular malformation
Endocri- nologic	Congenital adrenal hypoplasia	Congenital adrenal hypoplasia	Congenital adrenal hypoplasia	-
Immunologic	-	PID Autoimmune enteropathy	PID Autoimmune enteropathy	-
Psychologic	Food aversion	Food aversion	Food aversion Neglect	-

FPE, food protein-induced enteropathy; FPIAP, food protein-induced allergic proctocolitis; FPIES, protein-induced enterocolitis syndrome; GERD, gastroesophageal reflux disease; NEC, necrotizing enterocolitis; PID, primary immunodeficiency; T1DM, type 1 diabetes mellitus; VEOIBD, very early-onset inflammatory bowel disease.

**Table 5 nutrients-12-02086-t005:** Food-specific dietary advice and food co-reactivity in non-IgE-mediated gastrointestinal food allergies.

	FPIES	FPE	FPIAP
Cow’s milk	1st choice: EHF (10–20% reactivity) AAF if failure of EHF Soy formula: 40–60% co-reactivity ^†^ Rice formula: co-reactivity unknown May tolerate baked cow’s milk	First choice: EHF (20% reactivity) AAF if failure of EHF Soy formula: 10–30% co-reactivity ^†^	If BF: 1st choice: maternal elimination diet Alternative: EHF If on formula: 1st choice: EHF (10–20% reactivity) AAF if failure of EHF Soy formula: 10–30% co-reactivity ^†^
Soy	Cow’s milk: 40% co-reactivity ^†^	-	May be eliminated if no improvement with cow’s milk exclusion alone
Rice	Oats: 25–40% co-reactivity Wheat: 0–5% co-reactivity Corn: 1%: co-reactivity	-	-
Chicken	Avoid all poultry *: up to 40% co-reactivity	-	-
Egg	May tolerate baked eggs	-	May be eliminated if no improvement with cow’s milk/soy exclusion alone
Fish	Avoid all fish *: up to 80% co-reactivity between white & red fish Shellfish: 50% co-reactivity	-	-
Maternal elimination diet in BF	No, unless symptomatic	Unknown	Yes

^†^ May vary in different countries; * Unless already tolerated. In select cases, determination of cross-reactivity can be attempted with an OFC; AAF, amino-acid formula; BF: breastfed; EHF, extensively hydrolyzed formula.

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
