# Peer review of "Non-IgE-Mediated Gastrointestinal Food Allergies in Children: An Update"

_nutrients, 2020, doi:10.3390/nu12072086_

Round 1

Reviewer 1 Report

I reviewed this paper from the perspective of a Nutritionist. Therefore, it was pleasing to see a conclusion recommending early introduction of allergenic foods into the diet. My experience with restrictive diets has not been positive. This is a major contribution to our understanding of pediatric food allergies, which are poorly understood. Having spent a lifetime in the field it would seem a disservice to impede its publication. The use of acronyms was a problem for me. This is not my area of research and these acronyms are not ones I ordinarily use. CM for cow's milk? There is great information here and my head was swimming with "What does this acronym stand for?" Please understand that this is intended to be a positive criticism. I want this paper published and understood!

Author Response

We thank you for your comments and for the positive response. We have modified the CM acronym as suggested in the text and in the tables/figures. 

Reviewer 2 Report

This is an excellent review about Non-IgE-mediated gastrointestinal food allergies. It is well organized and clearly written. I have no major criticisms; however, I would recommend:
1. a clear definition of the different forms of allergy highlighting how these conditions meet the definition. For this reviewer, an adult gastroenterologist, non-IgE-mediated gastrointestinal food allergies do not fulfill my "expectations" for a food allergy. For example, an "allergic-type response" to all solid food is an oddity. Is this a term of convenience or can the authors substantiate their definition of these conditions as allergy.

2. I am pleased that the authors included some information concerning the much more common entity of food intolerance; however, some additional information would be welcome. It is known that, also in patients with lactose intolerance, there is an increase in intra-epithelial lymphocytes and other inflammatory cells on colonic biopsy and also that there is an increase in inflammatory cytokine release in these patients after ingestion of lactose (e.g. Yang, J., et al. (2014). Aliment Pharmacol Ther 39(3): 302-311). The symptoms described by patients with LI can include nausea as well as abdominal pain and diarrhea. Is this condition also on the "spectrum" of allergic responses as defined by the authors or is it truly distinct?

Author Response

Response 1. Thank you for your comment. We have added the following definitions lines 60 to 67 in the classification and terminology section:

Adverse food reactions are divided in immune-mediated reactions (ie. food allergy) and non-immune mediated reactions (ie. food intolerances). The term food allergy is used to designate an immune-mediated adverse reaction to food proteins. This includes IgE-mediated food allergy (IgE-FA), mixed IgE and non IgE-mediated food allergy, and non-IgE-GI-FA (PMID: 28723552). On the other hand, adverse non-immune mediated reactions that are not classified as food allergy include malabsorption due to enzyme deficiency (ex: lactase deficiency), reaction to toxic contaminants (ex: scombroid poisoning), and pharmacologic food components (ex: caffeine) among others (PMID: 30263035), which are beyond the scope of this review”.

 In addition, allergic reactions to some solid foods (particularly in the context of FPIES) is meant to distinguish between those patients who react to cow’s milk and/or soy formula (liquids), which was historically the classical presentation of FPIES, and those who react to one or more specific solid foods such as rice, eggs, or fish, for example. We have clarified this in the text (lines 280-281), with examples of common solid foods listed thereafter (lines 281-283).

Response 2. We thank the author for this comment. Although very interesting, we have chosen not to discuss lactose intolerance as the pathomechanisms involved are predominantly non-immune mediated.